# A Preliminary Longitudinal Study on Infant-Directed Speech (IDS) Components in the First Year of Life

**DOI:** 10.3390/children10030413

**Published:** 2023-02-21

**Authors:** Flaviana Tenuta, Roberto Marcone, Elvira Graziano, Francesco Craig, Luciano Romito, Angela Costabile

**Affiliations:** 1Department of Cultures, Education and Society, University of Calabria, 87036 Arcavacata, Italy; 2Department of Psychology, University of Campania “Luigi Vanvitelli”, 81100 Caserta, Italy; 3Unit for Severe Disabilities in Developmental Age and Young Adults, Scientific Institute IRCCS Eugenio Medea, 72100 Brindisi, Italy

**Keywords:** infant directed speech, dyadic interaction, emotional regulation

## Abstract

Infant-directed speech (IDS) is an essential factor of mother–infant interaction and facilitates early language processing and word learning during dyadic interactions. This study aimed to longitudinally investigate emotional and prosodic components of IDS in the first year of life by analyzing children’s responses to the different prosodic trends that mothers use during the observation of mother–child interactions. Seventeen mothers and infants were recruited for this study during their first hospitalization. The study involved observing communication changes in face-to-face interactions between the mother and child at three, six, and nine months after the child’s birth. We analyzed the relationship between gaze direction, smiling, speech speed, and clarity. The results showed that the IDS differs in production when compared to the age of the child; at only nine months, there is high intensity. The same is evident from the results related to the elocution velocity. The verbal sensitivity of the mother and the ability to tune in to the affective states of the child, especially at nine months, can predict the child’s understanding of future language.

## 1. Introduction

The caregiver’s sensitivity to responding appropriately to the child plays a central role in the development of relational reciprocity and mutual regulation [1,2,3]. During mother–child interactions, both partners exhibit mutual adaptations in gaze, posture, and play, leading to co-regulation [4]; both individuals actively participate and have an impact on each other [5]. Recently, there has been a focus on the specific linguistic features used by mothers when speaking to their children, leading to the recognition of a unique code known as infant-directed speech (IDS) [6]. IDS refers to the specific vocal pitch commonly used by parents during interactions with their infants. Infants show a preference for IDS over adult-directed speech (ADS) [7,8,9], and it differs from the natural speech used in conversations with adults due to specific prosodic, lexical, syntactic, and functional differences [10,11,12]. Initially, newborns may show a preference for their mother’s ADS over IDS [13], which may be a result of their prenatal exposure to mostly ADS during their mother’s speech. Research suggests that infants between six and thirteen months have different listening preferences compared to younger or older children. For instance, six-month-old infants show greater sensitivity to differences in verbal repetition between IDS and ADS compared to differences in prosody when compared to other age groups [14]. Furthermore, research has revealed that the preference of seven-to-nine-month-old infants can be influenced by acoustic–phonetic differences between IDS and ADS that extend beyond prosody.

Although the development of preference may not follow a linear path, children typically exhibit a preference for IDS due to an innate emotional connection to the melodic and rhythmic aspects of their mother’s speech. IDS is seen as a social signal that directs the infant’s attention toward referential targets. Infants at 6 months old are more likely to focus on an object that an adult is looking at and referring to with IDS (infant-directed speech) than with ADS (adult-directed speech) according to research [15]. IDS seems to be a component of a maternal interaction style that nurtures the emotional and verbal communication abilities of the growing infant. This style of communication also affects the child’s attention during these two-way interactions [16]. The caregiver utilizes guide questions to create reliable points of shared focus and attention. Once a shared focus of attention has been established between the caregiver and the infant, the caregiver can guide the infant’s attention either by aligning with the object to which the infant is already paying attention (following in) or by directing the infant’s attention to a new object of interest to the caregiver (directives) [17,18]. The importance of smiling as a means of communication in dyadic exchange is emphasized. Several researchers have shown the appearance of the “social” smile between 8 and 12 weeks and further noted that children become more intentionally communicative in this period. Children look at the mother and smile, and these behaviors tend to coexist when the mother offers objects to the children [19]. Previously, in object play, the mother was the one attuned to the child’s emotional experience; however, between 9 and 12 months, the frequency of instances where the child initiates affective attunement increased. Venezia and colleagues [20] studied positive affection communication in the context of object play and discovered that between the ages of 8 and 12 months, the child tends to significantly increase the behavioral pattern of “smiling, looking at an object, then turning their gaze to the adult’s face while maintaining the smile” [21].

Thus, IDS should be regarded as an emotional form of speech. Several studies highlight the impact of emotion on both IDS production and its effects, particularly with regard to linguistic and prosodic characteristics [22]. Previous studies described the specific linguistic characteristics of IDS, reporting that utterances are shorter, articulated clauses are rare, and parents use a lower proportion of different verbs, function, and content words and more diminutives, subject pronouns, and onomatopoeic sounds [23,24]. Concerning the prosodic and acoustic features of IDS, several parameters distinguish IDS from ADS, such as intensity, duration, velocity, and prosody, but among these, the parameters that seem to better draw infant interest relate to the fundamental frequency (F0) of the voice [25]. F0 represents the rate of vibrations of the vocal cords within the larynx and is acoustically perceived as the pitch of the voice. The acoustic analyses of F0 are positively associated with subjective judgments of emotion [26]. Thus, prosody (which is linked with F0 values and contours) reveals affective quantity and quality. The literature on infants’ perception of facial and vocal expressions indicates that infants’ recognition of affective expressions relies first on the multimodally presented information and then on the recognition of vocal expressions and finally on facial expressions [27].

Therefore, the emotional and prosodic components of IDS seem to be interconnected and play an important role in infant and child development. The IDS in the language environment is positively related to children’s language outcomes, such as vocabulary size [28,29,30], and it is increasingly being understood as a key predictor of both cognitive and early social skills. Considering the role of IDS in infant development, there is a growing need to longitudinally investigate communication development in the first year of life by analyzing children’s responses to the different prosodic trends that mothers use and demonstrating how children are more sensitive to IDS than adult speech.

Our hypothesis was that the coordination and synchrony between mothers and infants at three, six, and nine months of age is crucial for the production and maintenance of IDS. Further, we hypothesized that infants would exhibit a preference for IDS and that this preference would have an impact on the emotional aspect.

Hence, the goals of the current study are as follows: (1) to examine the alterations in face-to-face dyadic communication at three, six, and nine months of the child’s life by analyzing the relationship between gaze direction, gaze duration, smiling, and interaction between the two participants; (2) to determine the nature of the (proto) inter-action by measuring the frequency of vocalizations and articulations during dyadic interactions.

## 2. Materials and Methods

### 2.1. Participants

In this study, seventeen mother–child pairs were recruited at the “Annunziata” Hospital in Cosenza, Southern Italy, during their first hospitalization. All participating mothers provided informed consent for their participation in the study and for the use of their data, which was approved by the ethics committee of the University of Calabria. This study was conducted in accordance with the ethical guidelines set by the Italian Psychology Association (AIP) and the American Psychological Association (APA). After obtaining informed consent, the mothers were contacted by telephone for additional information and to schedule the appointment.

The average age of the seventeen mothers was 30.85 (SD = 3.05; range 26–35). The socioeconomic level of the seventeen families was high, with an average score of 56.15 (SD = 11.07) on the Hollingstead scale (1975). There were eight male children (38.5%) and nine female children (61.5%) involved in the study. All children were born at full term (mean gestational age = 39.61 weeks; SD = 1.12; range 38–41), with an average birth weight of 3211 kg (SD = 0.412; range 2.55–4) and without any perinatal or postnatal complications.

The study was performed in accordance with the Declaration of Helsinki and was approved by the Ethics Committee of the University of Calabria (ARC_1564138347906, 24 July 2019).

### 2.2. Procedure and Materials

The observations were recorded using a one-way mirror and videotaped in five-minute intervals in a laboratory setting. The mother–child pairs were observed in the Educational Psychology Laboratory of the Department of Culture, Education, and Society at the University of Calabria. The laboratory was kept free of visual distractions by curtains that were drawn to obscure it. The observations of a child’s development were recorded at three-month intervals, starting at three months old and using a longitudinal drawing method. The environment was structured to include a collection of age-appropriate toys, including the following: (1) a soft, multi-colored ball, (2) a tiny bucket, (3) a compact sound keyboard, and (4) two plush toys with bright colors. The mother was instructed to place her child in a designated highchair and to take a seat on a stool facing the child while the highchair was placed on a sturdy floor surface. The mother was then encouraged to engage in unstructured play with her child using the provided toy set.

We evaluated the speed of elocution and articulation. Elocution velocity is the ratio of the number of syllables in an utterance, including pauses, to a given run or the time taken to pronounce the utterance itself (VdE = (n. syllables)/run). Articulation speed (VdA) is the ratio of the number of syllables of a given utterance, excluding pauses, to the run.

Attention was focused on the conversation and considered four fundamental aspects [31]: (1) the sequential organization of actions, (2) the turn-taking system, (3) word overlaps, and (4) repair mechanisms.

The observational focus is on taking turns and overlapping, thereby identifying which of the two actors abandoned or followed up an exchange. Therefore, the Code of Evaluation of Shift Alternation was used, which characterizes the activity of the mother and child as proposed by [32]. The code emphasizes the ideas of simultaneous occurrence and synchronized interactions during play sessions. The behaviors of both the mother and the child are recorded and coded at two separate points in time.

Regarding the behaviors of the children, the elements analyzed were smiles, unreciprocated smiles, overlaps, eye contact, and gaze at objects; maternal behaviors were considered the same elements with the addition of baby talk.

The video was transcribed and recorded on a checklist, and it was then coded by two trained observers. The training process for the observers involved acquiring competence in the coding system using video recordings that had been previously coded by an expert. The reliability of the two coders was assessed using Cohen’s K (1960) [33] method. This involved conducting a time sampling encoding and calculating a sequential Cohen K, corrected for a one-second deviation, to determine the level of agreement over time. The K index was found to be, on average, above 0.70, indicating a high level of accuracy in both the coders’ interpretation of the code and their ability to accurately record the start and end of each interactive session with a maximum deviation of one second [34].

For each shift, the individual moves were analyzed following the Pra.Ti.D. scheme, which allows the analysis and annotation of Italian dialogical texts [35].

Sound Forge version 4, PRAAT 6.0.15, and ELAN 5.0.0–beta were used to analyze conversational moves; the latter allowed a conversational analysis using audio/video recordings following the Pra.Ti.D. scheme (see Figure 1).

Il Pra.Ti.D. (see Figure 1) proposes a three-tier division of dialogical moves. The first level includes the moves of “opening” or “closing” or some defined as “autonomous”; that is, those that are not conditioned or condition the development of interaction while having a communicative function. At the second level, the opening and closing moves are divided into five macro-categories depending on the communicative intention of the dialogical act (third level).

(1)Null describes a move of alignment between the participants that serves to open the dialogical exchange.(2)Influencing provides participatory feedback to the action of the interlocutor in order to complete the task. This includes moves such as the Action Directive (in which the proposer instructs their interlocutor), the Open Options (in which one of the two participants in the communication exchange gives directions to continue the conversation), and the Explains (an explanation and direct description of what the subject of the exchange is).(3)Question requires communicative feedback relative to the interlocutor compared to the previous move. These include polar and non-polar questions (Query Y/N and Query Wh), requests for information (Info Request), confirmation requests (Check), and an alignment by the subjects who are interacting (Align).(4)Understanding underlines the reception of the message by the interlocutor via acknowledgment signals, anticipatory completion (Continue), a time take-up using repetition (Repeat_Rephrase), a phatic expression (Phatic), a manifestation of understanding and solution of the game (Over), and its misunderstanding (Not_Ready).(5)Answer realizes a communicative contribution from the requested interlocutor both explicitly and implicitly. As a result, there will be explicit answers to the questions previously asked (Reply Y/N; Reply Wh), clarifying answers to what has been said (Clarify), objections (Objects) or corrections (Correct), and answers that indicate doubts and uncertainties (Hold).

The parameters used concern durations, speech speed, and prosody.

### 2.3. Data Analysis

Statistical analyses were performed using ELAN 5.0.0-beta, Generalized Sequential Querier (GSEQ 5.1.23), and STATISTIC 10 (©StatSoft, 1984–2011).

Descriptive analyses of the intensity of the mothers’ speech were performed; these focused on co-occurrence and dyadic synchronicity during game sessions, considering at two distinct moments both the behaviors of the mother and those of the child at three, six, and nine months of age.

To confirm the presence of a “Growth” factor, ANOVA for repeated measurements and *t*-tests for dependent samples were conducted, with the factor “Growth” (three, six, and nine months) as an independent variable and the behaviors of the mother (baby talk, smiles, unreciprocated smiles, overlaps, eye contact, and looking at the object) and the child (smiles, unreciprocated smiles, overlays, eye contact, and looking at the object) as dependent variables for all occurrences and durations.

Correlational analyses were performed for variables related to dialogic interchanges.

## 3. Results

### 3.1. Speech Analysis

The talk of mothers (see Figure 2), both in the different recording sessions (intra-mom) and between the recordings (inter-mom), had similar acoustic parameters. The fundamental frequency or first harmonic, which perceptually corresponds to the height (or tone) of the sound produced, of a conversational speech is on average between 70 and 150 Hz for a male voice, between 150 and 250 Hz for a female voice, and between 250 and 350 Hz for a child’s voice [36]. The values found for mothers in this study were all between 150 and 450 Hz (M = 302; D.S. = 81.11), highlighting how the average fundamental frequency values (F0) of the mothers exceeded the standard defined threshold values of conversational speech.

The value of the intensity must be differentiated into whispered production (when it is less than 40 dB) and normal speech (on average between 40 and 45 dB with peaks of 60 dB in the moments preceding a quarrel and 70–72 dB during an excited quarrel). These values are the result of specific research on the conversational analysis, which is present in a database of entries kept in the Phonetics Laboratory of the University of Calabria; these entries belong to the same linguistic area as the mothers who participated in this study. Our results show that the whispered voice had average intensity values below 40 dB, while in normal speech, the intensity was between 50 and 70 dB, therefore exceeding the normal threshold. The qualitative observations of conversational exchanges in video recordings clarified how higher peaks were emitted by the mother to point out or emphasize words.

Moreover, considering the duration of the utterance (ordinary speech has a speed of 2.5 syllables per second during the speech), the results highlight how the average speed recorded by our mothers was 1.5 syllables per second (SD = 0.8), emphasizing how the elocution and therefore the articulation of individual sounds were particularly slowed. The average duration of the syllables in the normal speech of our mothers in short statements was 0.20 s, while in long statements, the duration increased to 0.85 syllables per second (Figure 2).

### 3.2. Behavior in the Communication Exchange (Code of the Evaluation of Shift Alternation)

A second analysis considered the behaviors that are involved in communication, such as smiling, eye contact, unreciprocated smiles, objects, and overlaps. Maternal smiles and baby smiles were also expressed in average durations in seconds (Table 1).

The results of ANOVAs for repeated measurements show particularly interesting data; if there was no difference in the occurrences of smiles emitted by the mother (F(2.32) = 3207; n.s.) or by the child (F(2.32) = 2175; n.s.) at three, six, and nine months, then there was a significant decrease in average maternal smile times as the child’s age increased (F(2.32) = 60,677; *p* < 0.001) and, conversely, an increase in average smile times by the child as their age increases (F(2.32) = 11,661; *p* < 0.001) (Figure 3). When compared with the occurrences, this could tell us more about the quality of the exchange than the volume of exchange and how the dyadic uniqueness comes to an end depending on the growth of the child (Figure 3).

As for the average duration of smiles, it is intriguing to observe that at the age of three months, the mothers greatly extended the average time of elicitation of the smile, while the child kept the smile for a very short time. Already at six months and even more at nine months, the baby began to lengthen their average smile times, therefore allowing the mother not to “exaggerate” when exhibiting such behavior so that their average smile times could gradually wane.

In line with the literature data, therefore, at three months, the mother did not increase the number of smiles she directed at the baby, but rather prolonged on average (exaggerating) the time of each single smile. At the same time, along with a reduction in speed and the lengthening of smiles or the duration of a single smile, one should also record the lengthening of syllables. Over time, a trend reversal increased the speed of eloquence, reduced the duration of a single smile, and increased the total number of smiles. This previous behavior no longer seemed necessary when the child began to keep their smile for significantly longer periods of time.

A comparison of the average occurrences of the smile behaviors of mothers and children shows that the dyads moved in unison in the transition from three to nine months (Figure 4).

When other variables were considered, on which we had only measured occurrences, there were no substantial variations in the average behaviors as the children grew older: Eye Contact (F(2.32) = 1921; n.s.), Look at the Object (F(2.32) = 0.764; n.s.), and Smiles not Reciprocated (F(2.32) = 0.424; n.s.).

Consequently, there was a significant increase in the average occurrences of overlaps as the children’s ages increased (F(2.32) = 4739; *p* < 0.05); in particular, the post hoc analysis exhibited a significant increase in overlaps at the age of nine months, while the average number of overlaps at three and six months remained substantially unchanged (Figure 5).

### 3.3. Dialogical Moves (Pra.Ti.D.)

Regarding the codification of Pra.Ti.D., the dialogical moves referring to the Influencing category were initially analyzed, that is, those behaviors that denote participatory feedback from the interlocutor (Table 2).

The results of the *t*-tests for dependent samples underlined a significant difference in the average occurrences of Action Directives implemented by the mother from three to nine months: The more the child grew, the more the mother increased the number of instructions she gave to her child. This increase was statistically significant in the comparison between three and nine months (t(16) = 3706; *p* < 0.05) (Figure 6).

Open Option and Explain behaviors remained unchanged between three, six, and nine months.

Thus, regarding the Influencing macro-category, it can be said that, as the child’s age increased, the mother provided more instructions without changing the number of behaviors aimed at providing further explanations or instructions to complete the task.

Table 2 summarizes the descriptive results regarding the macro-category Question, that is, the implementation of communicative feedback addressed to the interlocutor with respect to the previous dialogical move.

The results of the *t*-test show a significant difference between three and six months in the number of confirmation requests (Check) (t(16) = 2161; *p* < 0.05), indicating that at three months, there was a greater need for continuous checks to maintain the dyadic tone. However, we can say that during all nine months, mothers tended to remain consistent in the average number of occurrences of such behavior.

A different issue regarding the mother’s implementation of communicative feedback via different types of questions is that at three months, the mother implemented significantly higher communicative feedback than she did at six and nine months (t(16) = 7698; *p* < 0.001 and t(16) = 3228; *p* < 0.01, respectively), validating how mothers always need more confirmation at three months, whereas as early as the sixth month, this need decreased significantly. Similarly, the implementation of communicative feedback by the mother using yes/no questions appeared to be a necessary behavior to a greater extent at three months, when compared to the subsequent steps, and the results of the *t*-tests underlined a statistical significance between three and six months (t(16) = 2568; *p* < 0.05) and three and nine months (t(16) = 2455; *p* < 0.05) but not between six and nine months (t(16) = 1027; n.s.). There was increased growth in Info Request behaviors, and the results of the *t*-tests underlined that the peak was at nine months (*t*-test six-nine months: t(16) = 2374; *p* < 0.05). It was clear that mothers implemented “simple” request feedback behaviors at three months and then moved on to real requests for information at nine months.

Finally, as their children grew up, mothers seemed to need fewer alignment behaviors (Align), which implicitly highlighted a gradual increase in children’s attention skills. The results of the *t*-tests show significant growth from three to six months (t(16) = 5095; *p* < 0.001), six to nine months (t(16) = 4.217; *p* < 0.001), and consequently, in the comparison of three and nine months (t(16) = 7265; *p* < 0.001). Figure 7 summarizes these considerations.

Regarding the Question macro-category, it can be said that as the child’s age grew, mothers were less committed to drawing the child’s attention and proceeding with continuous alignments. Despite this, mothers continued to verify that the exchange continued, although after three months, the need to continuously ask the child verification questions before proceeding with the task decreased significantly.

In the Understanding macro-category, the results of the analyses conducted do not show deviations in the average occurrences of Over behaviors, as well as for Acknowledgment behaviors, which do not show statistically significant differences although there was a tendency to increase. In this macro-category, there were significant decreases in the averages related to phatic expression (affirmation) phrases emitted by the child as they grew older. The results of the *t*-tests carried out emphasize a significant decrease in the second month of life, both compared to what happened at six months (t(16) = 2608; *p* < 0.05) and even more so in the comparison of three and nine months (t(16) = 5800; *p* < 0.001). Similarly, if the child’s phonetic phrases decreased significantly at nine months, then the reductions in responses of non-understanding by the child at the same age were very evident (3 vs. 9 months: t(16) = 3429; *p* < 0.01) (Table 2 and Figure 7).

In summary, for the Understanding macro-category, we verified that as age increased, the child exhibited less fatigue and fewer episodes of misunderstanding. Positive behaviors did not undergo a statistically significant increase, either regarding the signals of agreement (Acknowledgment) or the solution of the game (Over).

The results of the *t*-tests conducted for the behavioral interchanges that make up the Answer macro-category (Reply Y/N, Reply Wh, and Clarify) show that as the child grew older, the clarification responses to the questions previously asked by the mother significantly decreased. At nine months, the average number of clarification responses was statistically lower than the average six-month occurrences (t(17) = 2496; *p* < 0.05) and even more so at three months (t(17) = 4642; *p* < 0.001). Moreover, after three months, non-argumentative responses decreased significantly (t(17) = 2425; *p* < 0.05) to align between six and nine months (t(17) = 0.160; n.s.). At three months, a proto-communication system was established with continuous calls from both actors to keep attention high. At six months, this continuous need decreased significantly (three vs. six months: t(17) = 6542; *p* < 0.001; three vs. nine months: t(17) = 4297; *p* < 0.001; six vs. nine months: t(17) = 0.523; n.s.), possibly due to the child’s increased skills. Remarkably, dyadic symmetry did not require learning but was dynamically self-structural in a natural and innate way (Table 2 and Figure 7).

The detection of autonomous moves has made it possible to detect that as age increased, Comments decreased; in particular, there was a significant decrease in average Comment occurrences as early as the sixth month of life (three vs. six months: t(17) = 3602; *p* < 0.01), which then remained unchanged from the sixth to the ninth month (t(17) = 0.740; n.s.). There were no significant fluctuations in the Null and Self-talk categories (Table 2 and Figure 7).

### 3.4. Correlation Analysis

Correlational analyses were performed between the different categories of the Pra.Ti.D. code (Table 3), the results of which show that the action directive, that is, the instructions that the proposer gives to his interlocutor, correlated significantly with a manifest understanding or solution of the game (Over) only at nine months (r = 0.65; *p* < 0.01) but not at three and six months (r = −0.05 and r = 0.24, respectively). The Open Options, that is, the instructions that the proposer provides in order to continue the communicative exchange, correlated significantly with a manifest understanding of the solution of the game (Over) both at three and six months (r = 0.73; *p* < 0.001 and r = 0.55; *p* < 0.05, respectively). The explanation or direct description of the subject of the communicative exchange correlated significantly with a manifest understanding or solution of the game (Over) at three months (r = 0.55; *p* < 0.05) but not later (six months: r = 0.38; nine months r = 0.18). The requests for feedback during the communication exchange correlated significantly both with the correct reception of the message and with an important communication contribution from the interlocutor. In particular, Open Questions (Query Wh) related significantly to three months with the signals of agreement by the interlocutor (Acknowledgement, r = 0.65; *p* < 0.01), the phatic expressions (r = 0.76; *p* < 0.001), the Clarifying Answers (r = 0.62; *p* < 0.01), and the Yes/No Answers (r = 0.62; *p* < 0.01). Open Questions were also correlated at nine months with Open Answers (r = 0.64; *p* < 0.01) and Yes/No Answers (r = 0.60; *p* < 0.05). Yes/No Questions, on the other hand, related to three months both with the Signals of Agreement (r = 0.51; *p* < 0.05) and with the phatic signals (affirmation) (r = 0.63; *p* < 0.01) and correlated with both three, six, and nine months with Open Responses (three months: r = 0.78; *p* < 0.001; six months: r = 0.59; *p* < 0.05; nine months: r = 0.68; *p* < 0.01). In addition, at three months, the Confirmation Requests by the first interlocutor correlated significantly with Clarifying Answers (r = 0.87; *p* < 0.001), which was not evident in the following months (six months: r = −0.12; nine months: r = 0.39). Request for Information correlated significantly with a three- and six-month Game Understanding and Solution Event (three months: r = 0.51; *p* < 0.05; six months: r = 0.62; *p* < 0.01) but not at nine months (r = 0.38), which correlated with the following signal: phatic expressions (affirmation) (r = 0.49; *p* < 0.05).

## 4. Discussion

The significance of communication and interaction between mothers and children is emphasized by this study. The circular dyadic interaction observed during the study is a crucial factor in the development of the child and the recognition of the mother’s role. From the earliest stages of life, children possess communication abilities and actively participate in interactions with their mothers [37]. They exhibit increased positive expressions toward their mother’s face when they are supported, and their expressive behaviors are coordinated. The significance of maternal sensitivity and symmetry in fulfilling a regulating function for the child’s emotional organization is emphasized by the bond formed in the dyad during communication exchanges [38,39,40]. Our research underscores how mothers participated in a dyadic exchange with their children, boosting the frequency and average length of communication behaviors of intersubjectivity as the child grew. The findings reveal that the IDS varies in its production when compared to the child’s age; at nine months old, high-intensity IDS was observed. The same outcome emerges from the dates related to the speed of the speech: Speech is slow and repetitive, with peaks in intensity and a constant frequency value. The verbal sensitivity of the mother, meaning the ability to tune in to the affective states of the child, especially at nine months, can predict the understanding of future language. A recent study [41] reported similar results that from an early age, newborns actively take part in their speech. From an evolutionary perspective, these results indicate that children already play a significant and active role in eliciting and, therefore, supporting the caregiver’s response at the age of three months. Mothers adjust their tone based on the emotional expressions they perceive from children, and it has been demonstrated that children, in turn, modify the quality of their vocalizations in response to adults.

The general linguistic and paralinguistic characteristics of IDS have been described in several previous studies. When compared to ADS, IDS is characterized by a shorter tone [42], which is linguistically simpler, with redundant expressions. Voice interactions during the early months of life seem to be bilateral and mutually regulated [43].

The act of smiling appears to foster a functional reciprocal exchange. Smiling often arises during the shared enjoyment of social play, and in these circumstances, it is often the child who mirrors the adult’s smile [36]. At this stage, children start to show a preference for interpersonal games featuring speed, variations in rhythm, strong emotions (such as smiling at their mother) and exploring the external world by attempting to touch and grasp objects. Our results emphasize how a smile during the early months of life is utilized as a response to stimulation and later evolves into a fundamental component of the interaction experience with the mother.

According to Bates and Dick’s [44] observations, children first learn to use tone and then words. If at six months the child produces only the sounds contained in the language he hears, around 9 months, babbling is no longer used exclusively to produce random sounds but is used to communicate. At this age, the child while reaching out toward the object, looks at the mother; therefore, the act of request is no longer similar to that in the previous period, of the instrumental, but it now takes the form of intentional reporting. In general, a positive commitment will allow a successful transition to joint attention, which is fundamental for the communication and production of communication gestures. The IDS can be widely classified as referential or regulatory, and mothers use these two forms of language selectively. Our results show that the more the child grows, the more the mother increases the number of instructions to continue the conversation. Previous research has shown that when children indicate objects, mothers respond with the names of the objects [45].

What facilitates mother–child communication is the intonation of the maternal voice. Children are more sensitive to baby talk sequences than adult speech. Assessing the speed of the mother’s elocution and articulation is useful for demonstrating how the more the mother stimulates and speaks to the child, the more the child will respond with movements, gestures, vocalizations, and smiles. Infants show a specific preference for the expression of positive emotions in communication with caregivers [46]. As children grow older, their communication becomes increasingly intentional and social. During the first two years of life, children progressively direct their vocalizations toward objects and individuals, indicating a rising “desire to learn” [47].

This study stands out from previous research due to the use of the Pra.ti.D annotative system. This system breaks down and classifies the successive dialogical acts in the dyadic relationship, placing a strong emphasis on the connection with emotional elements and evaluating the alignment and coordination between the mother and baby.

The present study utilized a longitudinal approach to monitor the progression of communication development in children undergoing typical development. It would be fascinating to investigate the correlation between IDS and cognitive development, as well as how communication difficulties, such as early signs of autism spectrum disorder, may impact this process and affect children’s growth.

This study has the major drawback of having a limited number of subjects, but its longitudinal design and sequential data analysis methodology enabled the understanding of the interactional changes between mother and child during the first year of life and allowed the observation of turn-taking interactions in a circular manner from the early months of life.

## 5. Conclusions

In summary, this study emphasizes the significance of the initial communicative interaction between mother and child. The dyadic interaction was demonstrated to be a crucial factor for the child’s growth and for the recognition of the mother’s role. What appears to be most conducive to a functional reciprocal exchange is the smile. The emotional regulation processes that shape the interaction between mother and child during the first year of life play a critical role in shaping their social and emotional development. The results of this study underscore the significance of dyadic communication between mothers and children. As the child grows, the frequency and nature of communication behaviors change, and mothers adapt their intonation and duration of speech (IDS) accordingly. These continuous adaptations foster positive interactions and have a positive impact on both the emotional and cognitive development of the child. Language plays an important role in this communication, but this study highlights that it is not the only means of interaction and that other behaviors, such as smiling, also play a crucial role in shaping the dyadic relationship.

## Figures and Tables

**Figure 1 children-10-00413-f001:**
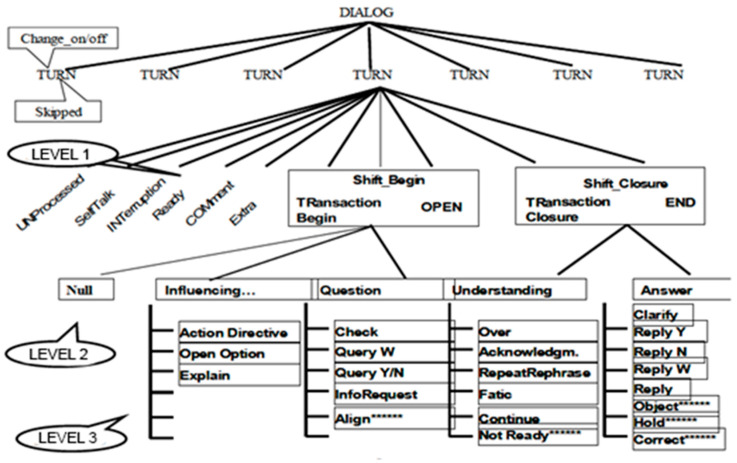
Graphic representation of the Pra.Ti.D model (Castagneto, 2012).

**Figure 2 children-10-00413-f002:**
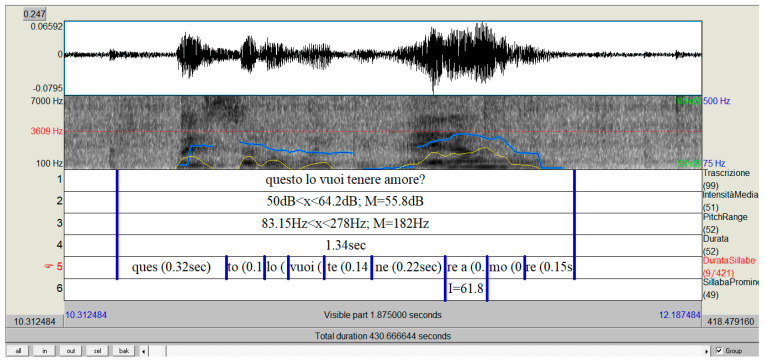
Oscillogram, sonogram, and spectrogram of the statement in which the intensity curve (in yellow) and that of the fundamental frequency (in blue) are visible; below is a series of TextGrids used for labeling.

**Figure 3 children-10-00413-f003:**
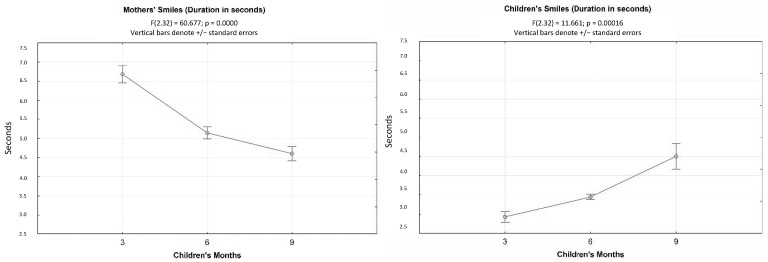
ANOVAs for repeated measures of mean. The duration of maternal smiles and infant smiles at three, six, and nine months.

**Figure 4 children-10-00413-f004:**
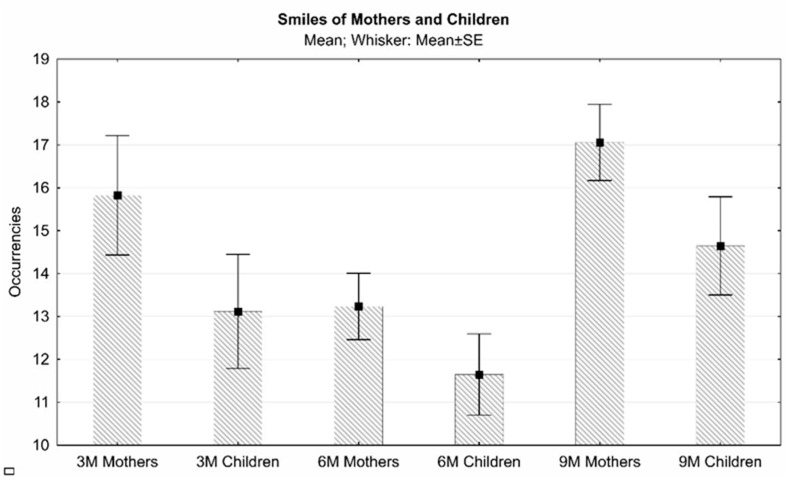
Mean and standard error of occurrences of maternal and infant smiles at three, six, and nine months.

**Figure 5 children-10-00413-f005:**
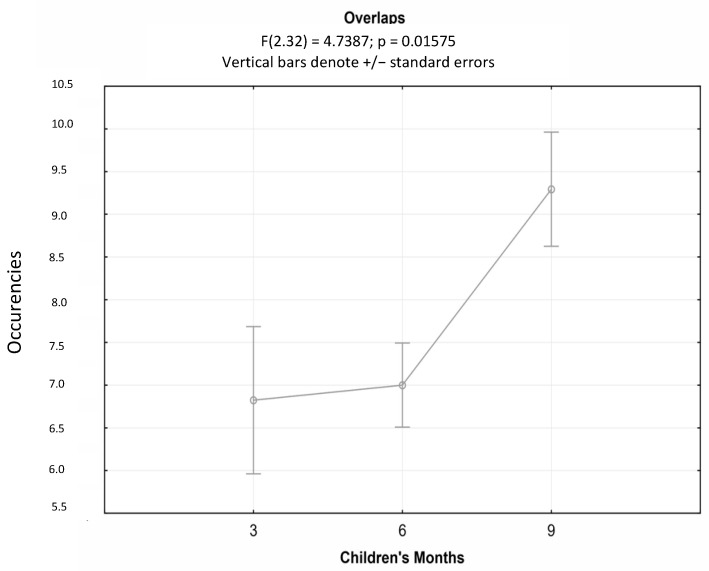
ANOVAs for repeated measures of the average occurrences of overlaps at three, six, and nine months.

**Figure 6 children-10-00413-f006:**
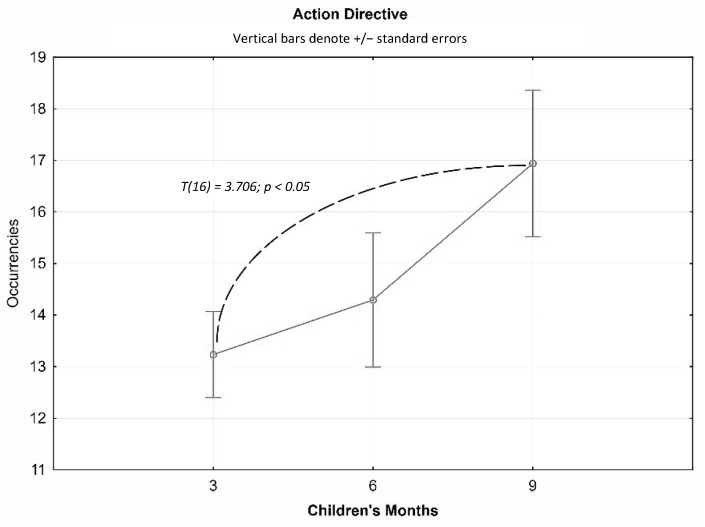
*T*-test average occurrences of the Action Directive at three, six, and nine months.

**Figure 7 children-10-00413-f007:**
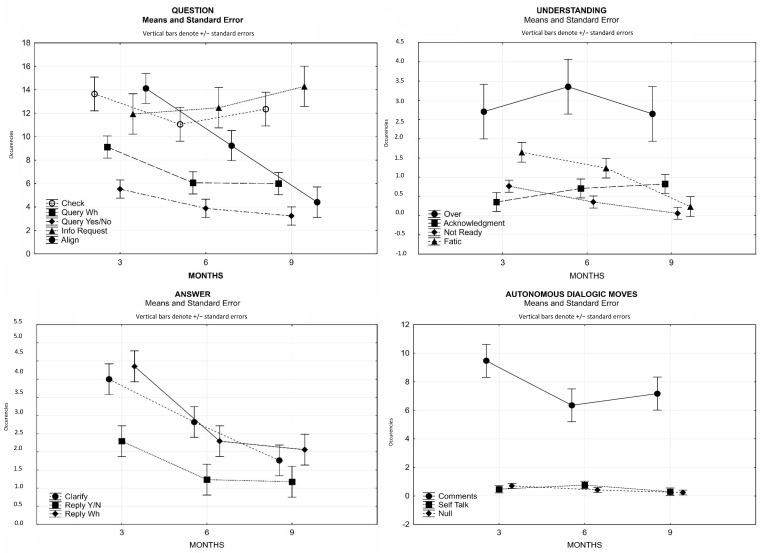
Means and standard error of Pra.Ti.D codifications at three, six, and nine months.

**Table 1 children-10-00413-t001:** Descriptive statistics at three, six, and nine months of the occurrences and duration in seconds of maternal and infant smiles.

Variabile	Months	M.	D.S.	Range
Maternal Smile (occurrences)	3	15.82	5.74	7–14
	6	13.24	3.19	7–18
	9	17.06	3.67	10–23
Child Smile (occurrences)	3	13.12	5.48	5–24
	6	11.65	3.90	7–18
	9	14.65	4.72	6–23
Maternal smile (durations)	3	6.68	0.93	5.50–8.40
	6	5.15	0.67	4.60–6.78
	9	4.60	0.79	3.70–5.90
Child Smile (durations)	3	2.93	0.59	2.30–4.30
	6	3.45	0.30	3.00–3.87
	9	4.51	1.38	2.87–6.65
Contact (occurrences)	3	7.59	5.05	1–19
	6	6.77	2.28	3–10
	9	9.18	2.77	4–14
Look at the object (occurrences)	3	5.53	2.70	1–11
	6	4.88	2.62	1–9
	9	5.94	2.88	2–11
Unrequited smile (occurrences)	3	2.24	2.49	0–8
	6	1.59	2.50	0–8
	9	6.82	2.74	0–9
Overlaps (occurrences)	3	6.82	3.56	1–14
	6	7.00	2.03	4–10
	9	9.24	2.76	2–14

**Table 2 children-10-00413-t002:** Codification of Pra.Ti.D.

Variable	Month	Mean	SD	Range
**Influencing**				
Action Directive	3	13.23	3.44	4–19
	6	14.29	5.37	5–21
	9	16.94	5.87	7–25
Open Option	3	11.76	7.13	1–22
	6	10.47	6.62	0–23
	9	12.41	6.17	2–25
Explain	3	9.24	6.48	0–21
	6	9.12	4.73	1–20
	9	8.53	5.93	0–23
**Question**				
Query Y/N	3	5.53	2.90	0–11
	6	3.88	2.52	1–9
	9	3.24	4.01	0–13
Query Wh	3	9.12	3.20	5–15
	6	6.06	3.07	1–13
	9	6.00	5.09	1–15
Info Request	3	11.94	6.00	1–19
	6	12.47	7.84	0–24
	9	14.29	7.29	1–25
Check	3	13.65	6.04	3–22
	6	11.06	4.72	4–18
	9	12.35	6.86	2–25
Align	3	14.12	6.88	4–25
	6	9.24	5.61	0–19
	9	4.41	2.50	0–9
**Understanding**				
Acknowledgment	3	0.35	1.06	0–4
	6	0.71	0.85	0–2
	9	0.83	1.13	0–3
Phatic	3	1.65	0.86	0–3
	6	1.24	1.56	0–5
	9	0.24	0.44	0–1
Over	3	2.71	2.23	0–8
	6	3.35	3.10	0–9
	9	2.65	3.35	0–13
Not Ready	3	0.76	0.90	0–3
	6	0.35	0.61	0–2
	9	0.06	0.24	0–1
**Answer**				
Reply Y/N	3	2.29	1.93	0–6
	6	1.24	1.44	0–5
	9	1.18	1.85	0–7
Reply Wh	3	4.35	1.84	1–8
	6	2.29	1.10	0–5
	9	2.06	2.14	0–5
Clarify	3	4.00	2.24	0–7
	6	2.82	1.42	0–5
	9	1.76	1.44	0–5
**Autonomous Moves**				
Comments	3	9.47	4.12	3–15
	6	6.35	4.20	0–17
	9	7.18	5.79	0–21
Self Talk	3	0.47	1.12	0–4
	6	0.76	1.39	0–5
	9	0.29	0.47	0–1
Null	3	0.71	0.99	0–3
	6	0.41	0.51	0–1
	9	0.24	0.44	0–1

*Note.* the name of five macro-categories are in bold.

**Table 3 children-10-00413-t003:** Significant correlations for alpha 0.05 between the variables of Pra.Ti.D. at (3), [6] and {9} months (N = 17).

(3 M) [6 M] {9 M}	OO	Ex	Ch	Q Wh	Q Y/N	IR	Al	Ov	Ac	Fa	Cl	R Y/N	R Wh	Co	Ho	Null
AD					(0.50 *)									(0.49 *)	(−0.48 *)	
			[0.52 *]											[0.51 *]		
	{0.65 **}		{0.66 **}					{0.65 **}								
OO		(0.72 ***)				(0.74 ***)	(0.61 **)	(0.73 ***)						(0.61 **)		
						[0.62 **]										
						{0.67 **}		{0.55 *}								
Ex			(0.60 *)			(0.69 **)		(0.55 *)						(0.53 *)		(0.60 *)

															[−0.57 *]	
Ch				(0.63 **)		(0.53 *)					(0.87 ***)			(0.45 *)		
						[0.63 **]		[0.51 *]								
					{0.70 **}											
QWh					(0.78 ***)				(0.65 **)	(0.76 ***)	(0.62 **)	(0.62 **)				
					[0.73 ***]											
					{0.56 *}		{−0.60 *}					{0.60 *}	{0.64 **}			
QY/N									(0.51 *)	(0.63 **)		(0.78 ***)				
												[0.59 *]				
												{0.68 **}				
IR							(0.76 ***)	(0.51 *)						(0.65 **)		
								[0.63 **]		[−0.54 *]				[0.68 **]		[0.50 *]
										{0.49 *}						
Al								(0.52 *)				(−0.53 *)		(0.71 **)		

										{0.59 *}						
Ov																

											{0.63 **}					
Ac										(0.56 *)		(0.62 **)				

												[−0.51 *]				
Ph											(0.52 *)					


RY/N																

													{0.52 *}			

**Note:** AD = Action Directive; OO = Open Option; Ex = Explain; Ch = Check; QWh = Query Wh; QY/N = Query Y/N; IR = Info Request; Al = Align; Ov = Over; Ac = Acknowledgment; Ph = Phatic; Cl = Clarify; RY/N = Reply Y/N; RWh = Reply Wh; Co = Correct; Ho = Hold. *****
*p* < 0.05; ******
*p* < 0.01; *******
*p* < 0.001.

## Data Availability

The data that support the findings of this study are available on request from the first or corresponding author.

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
