# Peer review of "A Preliminary Longitudinal Study on Infant-Directed Speech (IDS) Components in the First Year of Life"

_children, 2023, doi:10.3390/children10030413_

Round 1
Reviewer 1 Report
â—‹ Thanks for your effort for working on this paper.
â—‹ This study is meaningful in that it presents empirical evidence that IDS is expressed differently depending on the age of the child by longitudinally investigating the emotional and prosodic components of the IDS through the observation of mother-child interactions during the first year of life. This study is also meaningful in that it provides empirical evidence for the importance of the mother's role in child development.
â—‹ The research problem, purpose, methods, results are well documented. But minor revision needed.
â—‹ Please indicate in detail what differentiates this study from previous studies.
â—‹ Please add suggestions for follow-up research.
â—‹ Modify “Ovelaps” in Figure 5 to “Overlaps”
â—‹ Please replace outdated references with newer ones.
Author Response
Thank you for your helpful critique of our manuscript. We greatly appreciate the opportunity to submit a revision of the manuscript.
We respond to reviewers’ comments below in their original order.
Comment: This study is meaningful in that it presents empirical evidence that IDS is expressed differently depending on the age of the child by longitudinally investigating the emotional and prosodic components of the IDS through the observation of mother-child interactions during the first year of life. This study is also meaningful in that it provides empirical evidence for the importance of the mother's role in child development.
Author's response: We thank the reviewer for the positive comment.
Comment: Please indicate in detail what differentiates this study from previous studies.
Author's response: The Reviewer raises an important point. In the Discussion section, we provide a detailed account of what sets this study apart from previous studies.
Comment: Please add suggestions for follow-up research.
Author's response: Following the Reviewer’s suggestion, new suggestions for follow-up studies have been added in the Discussion section
Comment: Modify “Ovelaps” in Figure 5 to “Overlaps”
Author's response: The modification has been made
Comment: Please replace outdated references with newer ones.
Author's response: The modification has been made
My co-authors and I thank you for considering the revised manuscript.
Reviewer 2 Report
The study, despite not being totally original, and having few dyads, has its importance due to the longitudinal design. The introduction is well grounded, with relevant information for the understanding of the subject. The objectives are well described, but I missed the hypotheses of the study. The insertion of the hypotheses will be important. The method is also well described, with details of the instruments and the analyzed categories. The results were described objectively, with the use of tables and figures that help the readers to understand them. The discussion is aligned with the introduction and the results. In some moments I found that the English writing lacked some checking, nothing that compromises the general understanding of the study, but important to give fluency to the information, so I suggest a general revision of the English.
There is one part of the conclusion that I did not understand. I believe it is not possible to say that the study in question shows how maternal language impacts development (cause and effect) from these statistical analyses, but that the frequency of behaviors change as the child ages (lines 473-476). I also did not see in the results the comparison that the child prefers IDS to ADS. Conclusions should be restricted to the results of the study presented.
Some specific suggestions:
Phrase lines 142-143: "and the gaze at the projects" I didn't understand the meaning of this phrase. I think it's worth revising.
Figure 1 = change "livello" to level
Line 417 "intersubiectivity" correct to intersubjectivity.
Author Response
Thank you for your helpful critique of our manuscript. We greatly appreciate the opportunity to submit a revision of the manuscript.
We respond to reviewers’ comments below in their original order.
Comment: The study, despite not being totally original, and having few dyads, has its importance due to the longitudinal design. The introduction is well grounded, with relevant information for the understanding of the subject. The objectives are well described, but I missed the hypotheses of the study. The insertion of the hypotheses will be important. The method is also well described, with details of the instruments and the analyzed categories. The results were described objectively, with the use of tables and figures that help the readers to understand them. The discussion is aligned with the introduction and the results. In some moments I found that the English writing lacked some checking, nothing that compromises the general understanding of the study, but important to give fluency to the information, so I suggest a general revision of the English.
Author's response: The Reviewer raises an important point. Our hypothesis has been added to the introduction. In addition, we corrected the English using the Language Editing Services provided by the journal
Comment: There is one part of the conclusion that I did not understand. I believe it is not possible to say that the study in question shows how maternal language impacts development (cause and effect) from these statistical analyses, but that the frequency of behaviors change as the child ages (lines 473-476). I also did not see in the results the comparison that the child prefers IDS to ADS. Conclusions should be restricted to the results of the study presented.
Author's response: following the reviewer's suggestions, we have rewritten the conclusions
Some specific suggestions:
Comment: Phrase lines 142-143: "and the gaze at the projects" I didn't understand the meaning of this phrase. I think it's worth revising.
Author's response: The modification has been made
Comment: Figure 1 = change "livello" to level
Author's response: The modification has been made
Comment: Line 417 "intersubiectivity" correct to intersubjectivity.
Author's response: The modification has been made